# Long-Term Study of the Changes in Symptoms and Signs of Dry Eye Disease in Participants Non-Adherent to Treatment

**DOI:** 10.3390/life15111783

**Published:** 2025-11-20

**Authors:** Belen Sabucedo-Villamarin, Jacobo Garcia-Queiruga, Hugo Pena-Verdeal, María José Ginzo-Villamayor, Carlos Garcia-Resua, Maria J. Giraldez, Eva Yebra-Pimentel

**Affiliations:** 1GI-2092-Optometry, Departamento de Física Aplicada (Área de Optometría), Universidade de Santiago de Compostela, Campus Vida s/n, 15701 Santiago de Compostela, Galicia, Spain; belen.sabucedo@rai.usc.es (B.S.-V.); carlos.garcia.resua@usc.es (C.G.-R.);; 2AC-24-Optometry, Instituto de Investigación Sanitaria (IDIS), Travesía da Choupana S/N, 15706 Santiago de Compostela, Galicia, Spain; 3GI-1914-MODESTYA, Departamento de Estadística, Análisis Matemático y Optimización, Universidade de Santiago de Compostela, Campus Vida s/n, 15782 Santiago de Compostela, Galicia, Spain; mariajose.ginzo@usc.es

**Keywords:** DED natural course, OSDI questionnaire, tear film osmolarity, tear film stability, corneal staining, longitudinal study, maximum blink interval

## Abstract

Background: The purpose of this study was to assess differences in clinical symptoms and signs of DED in non-adherent to treatment patients to describe long-term disease progression. Methods: 120 patients previously diagnosed with Dry Eye Disease (DED) were contacted to undergo a second eye examination. The final included participants were classified into three groups based on when the second examination was scheduled: 4 years (Group 1; *n* = 33), 6 years (Group 2; *n* = 18) or 8 years (Group 3; *n* = 37) since the diagnostic visit. All included participants were classified as ‘non-adherent to DED treatment’, defined as patients who reported not following their prescribed DED therapy. In both examinations, Ocular Surface Disease Index (OSDI) questionnaire, tear film osmolarity, inter-eye osmolarity (osmolarity|OD-OS|), Fluorescein Break-Up Time (FBUT), Maximum Blink Interval (MBI) and corneal staining were evaluated. Results: OSDI score improved after 4 years of DED diagnosis (Group 1, mean difference close to 12 points, *p* < 0.001) and after 8 years (Group 3, mean difference of 9 points, *p* < 0.001), but remained stable after 6 years (Group 2, *p* = 0.328). Osmolarity worsened only after 6 years of DED diagnosis (Group 2, mean difference of 13.2 mOsm/L, *p* = 0.011), while osmolarity|OD–OS| showed no change (all *p* ≥ 0.231). FBUT values were stable across all groups (all *p* ≥ 0.265). MBI increased after 4 and 8 years of DED diagnosis (Groups 1 and 3, *p* ≤ 0.003), but not after 6 years (Group 2, *p* = 0.391). Corneal staining worsened after 8 years of DED diagnosis (Group 3, 0.55 points, *p* = 0.011), with no changes at 4 or 6 years (Groups 1 and 2, both *p* ≥ 0.318). Conclusions: In non-adherent DED patients, osmolarity|OD-OS| and tear film stability remain stable during the natural course of the disease, while ocular surface damage increases. However, the subjective symptomatology and the nociceptive blink reflex due to ocular discomfort decreased since the diagnostic visit.

## 1. Introduction

Dry Eye Disease (DED) was redefined by the Tear Film and Ocular Surface Society in the third Dry Eye Workshop (TFOS DEWS III) as a “a multifactorial, symptomatic disease characterized by a loss of homeostasis of the tear film and/or ocular surface, in which tear film instability and hyperosmolarity, ocular surface inflammation and damage, and neurosensory abnormalities are etiological factors” [1].

DED diagnosis is influenced by many factors such as age, sex, ethnicity or protocol method, among others, which complicate the study of the prevalence of the disease [2,3,4,5]. Although the TFOS DEWS II reported a prevalence of DED between 5 and 50% of the global population [2], these numbers may vary if the studies consider the diagnosis of DED based solely on signs, symptoms or both [3,5,6]. Nevertheless, the recent TFOS DEWS III report refined these estimates, indicating an overall global prevalence of approximately 11.6%, with values ranging from 2.7% in individuals aged 20–29 years to 30.1% in women over 80 years of age [1].

DED symptoms include ocular discomfort, often perceived by patients as pain, burning, or grittiness, together with visual disturbances such as fluctuating, poor or blurred vision [7]. Additionally, TFOS DEWS III highlights the multifactorial and symptomatic nature of the disease, where neurosensory abnormalities may contribute to the mismatch between signs and symptoms [1,7]. Within the signs, a high tear film osmolarity may be responsible for the pathophysiological process in which the disruption of ocular surface homeostasis occurs [7]. Therefore, hyperosmolarity is a key factor being a trigger and main contributor for the instability of the tear film, inflammation and damage of the ocular surface that induces the “Vicious Circle” of DED [8,9], because this inflammatory cascade reduces the expression of glycocalyx mucins and induces apoptosis of epithelial cells, decreases ocular surface wettability, increases tear film instability, and ultimately leads to hyperosmolarity [8].

This disease is generating a large increase in socio-economic costs in the healthcare systems: the costs are directly related to medical care spending, and indirectly to the cost attributed to the loss of productivity and impact on everyday life [2,10,11,12]. Therefore, DED is currently considered a public health problem [2,11]. In terms of other degenerative or age-related ocular conditions, it has been highly studied and characterised that parameters or signs get worse over time, such as the progressive crystalline opacification in cataracts, the increase in the intraocular pressure in glaucoma or the abnormal growth of retinal vessels in the age-related macular degeneration, worsening in absence of disease treatment [13,14,15]. However, the natural course of the DED condition remains uncertain, and the available longitudinal studies are also limited [2,16,17]. Practitioners advocate for long-term studies related to changes in both symptoms and signs over time in the evolution of DED to provide a better diagnosis, treatment and prognosis for DED patients in the future [2,16,17,18]. Due to the scarcity of research on this issue, the purpose of this study was to assess differences in clinical symptoms and signs of DED in non-adherent to DED treatment patients to describe long-term disease progression.

## 2. Materials and Methods

### 2.1. Sample and Study Design

The present protocol was devised as an observational prospective study divided into two different sessions. On each session, a battery of tests was conducted in accordance with the “TFOS DEWS II Diagnostic and Methodology Subcommittee report” which is supported and endorsed by TFOS DEWS III [1,7,9]: Ocular Surface Disease Index (OSDI), tear film osmolarity, fluorescein break-up time (FBUT), maximum blink interval (MBI) and corneal staining. Based on the hypothesis that severity should be independently assessed between eyes, or even utilised as a diagnostic indicator, both eyes of the participants were evaluated in the study [7,19,20,21]. All the participants gave their written informed consent to be included in the study. The present study adhered to the tenets of the Declaration of Helsinki and both stages of the study were approved by an Ethics Committee. In all sessions, the tests were conducted in the same order from the least to the most invasive [7]. The protocol was conducted under controlled environmental conditions of light, temperature (20–23 °C) and humidity (50–60%). Due to the time required to conduct the study, the procedures were not always performed by the same examiner, but the same devices were used for all measurements in both sessions. Moreover, all procedures in both sessions were video recorded and analysed by a second masked observer, who remained consistent throughout the analyses. All the data were masked by an alphanumeric code.

The initial assessments of patients (diagnostic visit, session 1), conducted to confirm DED diagnosis as a routine process in the Optometry Eye Clinic, was carried out between 2013 and 2017 (ethics committee approval code CEIG2013/360). All participants included in the study were initially referred by the institution’s Medical Service with a suspected diagnosis of DED for a comprehensive eye examination. The final diagnosis was established at the Optometry Eye Clinic based on the ocular surface evaluation and following the TFOS DEWS II diagnostic recommendations [7,9]. While all referred patients presented symptoms suggestive of DED, only those who also showed clinical sings during the evaluation were diagnosed with DED. Those participants with symptoms and signs consistent with the early stages of the disease, Mild to Mild-Moderate DED, were given a series of visual and ergonomic recommendations and tear substitutive instillation protocol, and a follow-up appointment in no more than 2 years, while patients with Severe DED were referred to their specialist doctors instead of the Centre’s clinic.

The second assessment (follow-up measurement, session 2), targeted patients who had not attended the centre for their check-up since the confirmation of the diagnosis. These participants were then contacted by the centre to voluntarily undergo a second new examination between 2021 and 2023 (ethics committee approval code USC-08/2021) [17]. During this contact process, the participants were asked if they followed the recommendations provided by eye care practitioners in the diagnostic visit. While some participants reported closely adhering to the recommendations and were subsequently scheduled for a follow-up examination, others acknowledged not following the recommendations consistently. Only those participants who admitted to non-adherence were included in the present study and categorised into three groups based on the time elapsed from the diagnostic session (4, 6 or 8 years). Non-adherent to DED treatment refers to patients who reported not following their prescribed therapy. Their data underwent analysis after the diagnosis of DED was reconfirmed [17].

Previous to the recruitment process, the sample size of each group was calculated based on the TFOS DEWS II Diagnostic Methodology report principle by using the software PS Power and Sample Size Calculations Version 3.1.2 (Copyright© by William D. Dupont and Walton D. Plummer, United States). The Standard Deviations (SDs) reported in the literature on the symptomatology status (OSDI score), tear film osmolarity, tear film FBUT, and fluorescein corneal staining were assumed to be 6.7, 4.8 mOsm/L, 2.9 s and 2, respectively. To have 80% power for a significance level of α = 0.05 (Type I error associated) to detect a minimal clinical difference of 7.3, 5 mOsm/L, 3 s and 2, respectively, between no pathological and pathological participants, the minimum number of participants required in each group was 11, 11, 12 and 10, respectively. The highest of those values was selected as the reference for determining the sample size (12 participants) to ensure a more robust study. To guarantee that this minimum statistically reliable sample size was obtained in each of the final recruitment groups, an adjustment of a 70% anticipated loss in follow-up led to a total sample size of 40 for each group during the initial contact:Group 1 (4-year evaluation)—33 participants (9 men, 24 women), with a mean age of 50.1 ± 10.4 years, were recruited from 40 participants who were initially offered a second examination after 4 years (1460 ± 21 days) of the diagnostic session.Group 2 (6-year evaluation)—18 participants (4 men, 14 women), with a mean age of 47.4 ± 11.3 years, were recruited from 40 participants who were initially offered a second examination after 6 years (2190 ± 21 days) of the diagnostic session.Group 3 (8-year evaluation)—37 participants (6 men, 31 women), with a mean age of 49.9 ± 8.8 years, were recruited from 40 participants who were initially offered a second examination after 8 years (2920 ± 21 days) of the diagnostic session.

None of the participants used any kind of medication, artificial tears or wore contact lenses during the assessed period, after the diagnostic visit [22]. Participants were excluded if they had a prior history of ocular infections, glaucoma, any ocular surgery, systemic disease, autoimmune disease, were pregnant or breast-feeding [22]. Figure 1 shows a CONSORT-style flowchart to facilitate understanding of how participants were included in the current study.

### 2.2. Evaluation Procedures

#### 2.2.1. Symptomatology Assessment

The symptomatology of DED was measured using the OSDI questionnaire [7,23,24]. The OSDI is a specific questionnaire composed of twelve questions asked concerning a 1-week recall period, with three subsets of questions related to visual function, ocular symptoms and environmental triggers. The participants completed an online self-administered questionnaire by scanning a QR code before the examination [7,23,24]. The scores obtained were evaluated by the examiner according to the standardised guidelines of 0 to 100 points, where higher scores represent greater disability [7,23,24].

#### 2.2.2. Tear Film Osmolarity

The tear film osmolarity was measured using the TearLab osmometer (TearLab Corp, San Diego, CA, USA) [25]. To perform the measurement, the participants were instructed to sit with their heads tilted and their eyes looking at the ceiling. The instrument probe was then placed on the lower tear meniscus until a beep was emitted, which indicates that the sample was correctly collected [25]. The device converted the electrical impedance of the tear film sample into osmolarity values (mOsm/L), and the results were displayed on the screen. The device measurement range extends from 275 to 400 mOsm/L. The same card lot was used during each session.

#### 2.2.3. Tear Film Stability Assessment

Tear film stability was assessed by measuring the FBUT and the MBI, both recorded by using the fluorescein function provided by Keratograph 5M (Oculus Optikgerate GmbH, Wetzlar, Germany) [26]. The participants were properly positioned in the instrument and instructed to look up to the ceiling to apply a saline-hydrated fluorescein strip to the lower bulbar conjunctiva. Participants were requested to blink several times to ensure the adequate mixing of the fluorescein dye [7,27,28]. They were then asked to look straight ahead to a red dot provided by the device [28]. FBUT and MBI procedures were performed three times, and once the videos were recorded and extracted, a second masked observer measured the FBUT and MBI using the VirtualDub64 v.1.10.4 software (Open software, link: https://sourceforge.net/projects/virtualdub/ (accessed on 6 March 2024)) [29]. The software provides the video recorded in frames, where one second contains 8 frames. The FBUT was defined as the interval between the last blink and the appearance of the first black spot in the tear film, while the MBI was defined as the maximum time the participant was able to keep his eye open [7,30].

#### 2.2.4. Corneal Staining

The ocular surface damage was assessed by the corneal staining measurement recorded by using the fluorescein function of the Keratograph 5M (Oculus Optikgerate GmbH, Wetzlar, Germany) [26]. The procedure was taken immediately before recording the tear FBUT and MBI. Participants were instructed to look to the centre of the device to evaluate the possible damage to the central cornea, and then look to the right, left, up and down to evaluate the entire cornea [28,31]. Once the video was recorded and the images of the entire cornea were extracted, the corneal staining was evaluated by another masked observer following the Oxford Scheme grades [32]. The Oxford Scheme goes from grade 0 to 5, which represents the severity of the damage in the ocular surface: where mild is grade 0 or 1, moderate is grade 2 or 3, and severe is grade 4 or 5 [32].

### 2.3. Statistical Analysis

Before the analysis of the differences between sessions, the differences between the measurements in the first session between the groups were assessed to establish whether an initial bias was present. This analysis was performed to ensure that any assumptions made when comparing the groups were not influenced by dissimilar initial distributions of characteristics (to mitigate the potential impact of any initial differences on subsequent analyses). Then, the differences between the diagnostic visit and second session values of all parameters were studied for each group separately. Hence, based on the data distribution, a paired-t test was used to assess the differences in the measurements between sessions in each group for OSDI, osmolarity and osmolarity|OD-OS| values, while the Wilcoxon test was used for FBUT, MBI and corneal staining values [33].

SPSS statistical software v.25.0 for Windows (SPSS Inc., Chicago, IL, USA) was used for data analyses. Statistical significance was set at a *p* < 0.05 for all the analyses [33]. Based on the study design, participants were grouped depending on when the second examination was scheduled; therefore, statistical analyses were performed based on those groups. First, the normal distribution of the data was checked for all parameters studied. Kolmogorov–Smirnov test was used to check the normality of the OSDI data for Groups 1 and 3, and for the FBUT, MBI and corneal staining data in Group 2. The Shapiro–Wilk test was used in all the other procedures. OSDI, osmolarity and osmolarity|OD-OS| values showed a normal distribution (all *p* > 0.05); while FBUT, MBI and corneal staining have not (all *p* ≤ 0.05). The inter-eye difference in osmolarity employed for the analysis was computed as the absolute difference between values obtained from both participants’ eyes (|OD-OS|), being named as osmolarity|OD-OS| [20,21].

## 3. Results

On the initial analysis of characteristics between groups in the diagnostic session, no statistically significant differences between groups were found in the OSDI, osmolarity, osmolarity|OD-OS|, FBUT, MBI and corneal staining measurements (ANOVA test, Kruskal–Wallis test or Fisher’s exact test; all *p* ≥ 0.085).

### 3.1. Natural Evolution of Symptomatology (OSDI Score) over Time

Table 1 shows the descriptive statistics values and the analysis of differences in the OSDI questionnaire for Groups 1, 2 and 3, respectively. OSDI scores showed a slight improvement after 4 years (Group 1, mean difference close to 7 points, paired-t test *p* < 0.001) and after 8 years of DED diagnosis (Group 3, mean difference of 6 points, paired-t test *p* < 0.001), but did not improve after 6 years of DED diagnosis (Group 2, score remained stable with a 6-point difference, paired-t test *p* = 0.058).

### 3.2. Natural Evaolution of Tear Film Osmolarity over Time

Table 2 shows the descriptive statistics values and the analysis of differences for the osmolarity of Groups 1, 2 and 3, respectively. Osmolarity values showed a worsening after 6 years of DED diagnosis (Group 2, mean difference of 8.44 mOsm/L, paired-t test *p* = 0.048), but did not change after 4 years (Group 1, mean difference of 8.59 mOsm/L, paired-t test *p* = 0.106) or 8 years of DED diagnosis (Group 3, mean difference of 4.34 mOsm/L, paired-t test *p* = 0.983). Also, osmolarity|OD-OS| did not show any change in any of the 3 groups (paired-t test, all *p* ≥ 0.231).

### 3.3. Natural Evolution of the Tear Film Stability over Time

Table 3 shows the descriptive statistics values and the analysis of differences in FBUT and MBI for Groups 1, 2 and 3, respectively. FBUT values did not show any change in any of the groups (Wilcoxon test, all *p* ≥ 0.133). However, MBI values showed an increasement after 4 years (Group 1, mean difference of 4.88 s, Wilcoxon test *p* < 0.001) and after 8 years of DED diagnosis (Group 3, mean difference of 4.56 s, Wilcoxon test *p* < 0.001), but did not show any change after 6 years of DED diagnosis (Group 2, mean difference of 2.73 s, Wilcoxon test *p* = 0.101).

### 3.4. Natural Evolution of the Corneal Staining over Time

Table 4 shows the descriptive statistics values and the analysis of differences in corneal staining for Groups 1, 2 and 3, respectively. The corneal staining values showed a worsening after 8 years of DED diagnosis (Group 3, mean difference of 0.50 points, Wilcoxon test *p* = 0.004), but did not show any change after 4 years (Group 1, mean difference of 0.31 points, Wilcoxon test *p* = 0.091) and after 6 years of DED diagnosis (Group 3, mean difference of 0.07 points, Wilcoxon test *p* = 0.593).

## 4. Discussion

DED has been described as a chronic and multifactorial disease accompanied by a variety of symptoms and signs [9]. It has high suffering rates, being even considered a modern global epidemic, and therefore, a public health concern. This condition causes a decrease in the quality of life of patients, who in severe cases must struggle with depression, anxiety and other mental health disorders due to their disabling symptoms [11,34,35]. Although strategies for DED prevention and treatment have been developed, studies of the natural course, including changes in symptoms and signs of the disease, remain limited [34]. Therefore, clinicians request more comprehensive studies which include changes in both symptoms and signs over time [2,16,17,18]. In the present study, changes in main diagnostic DED symptomatology and signs were analysed after 4, 6 and 8 years since the initial diagnosis of DED in non-adherent patients.

Regarding DED symptomatology, some authors found that patients with chronic pain and diseases such as DED, become more tolerant because their pain threshold is increased [36,37]. Xue et al. [38] which evaluated the ocular symptoms of dry eye before cataract surgery, found that the OSDI scores decrease gradually 6 months after surgery and end up stabilising. The DREAM© study found that there were no statistically significant differences in the symptomatology of DED in four different age groups (<50, 50–59, 60–69, ≥70 years old). These results suggest that once DED is diagnosed, the OSDI score may not be significantly affected by age, implying that the chronicity of this condition may not be severe in all cases [39]. Nevertheless, in the present study it was observed that the OSDI scores at the second session decreased significantly over time for all groups except Group 2, which remained stable (Table 1), indicating an improvement in subjective symptomatology over time. These results may support the hypothesis of an initial reduction in DED symptoms that is maintained over time.

Based on the clinical relevance of the tear film osmolarity, it is necessary to assess how this parameter changes during the natural course of DED. The present study shows osmolarity values between sessions (Table 2), with higher values in the 6-year follow-up group, while the 4- and 8-year follow-up groups remain stable within the expected values for the disease [9]. Although a statistically significant increase was detected in the 6-year group, this subgroup had the smallest sample size (*n* = 18, compared with *n* = 25 and *n* = 35 in the 4- and 8-year groups, respectively), which may have amplified random variability and contributed to the observed difference. Therefore, while previous reports suggest that osmolarity fluctuates throughout the day [40], the present analysis at 4, 6 and 8 years points that osmolarity stabilises over the natural course of the disease in individuals with DED. Furthermore, it is known that a greater inter-eye difference than 8 mOsm/L is considered a characteristic of DED diagnosis and in addition, this inter-eye difference increases as the severity of DED increases, while other diagnostic parameters such as FBUT and corneal staining showed no inter-eye differences diagnostic in terms of diagnostic ability [20,41]. Because of this, the osmolarity|OD-OS| was also assessed in the study and the results showed no statistical differences in any of the groups (Table 2). It is important to note that in the current study, all measurements were obtained using the same device model, following identical calibration procedures and under similar environmental conditions, thereby minimising potential diurnal, seasonal or lot-related effects. To the authors’ knowledge, there are no previous studies that have assessed the variance in this parameter over time.

Regarding tear film stability, the results of the present study have not shown any variation in FBUT values between sessions in any group (Table 3). Zao M et al. [39], although it primarily assesses how increasing age is associated with the signs and symptoms of DED, there were no significant differences in FBUT across age groups from baseline to 6 and 12 months. Surprisingly, no changes in FBUT were found through the natural course of the disease when tear film stability could theoretically decrease as a consequence of the ocular surface hyperosmolarity, which also did not change in any of the groups of the present study [8]. Although the use of tear substitutes can improve tear film stability [42,43], the results of the present study show that even without treatment of DED there is no worsening over time.

In addition, although the mean MBI values obtained in the first session were within the ranges described by several authors for patients with DED (Table 3) [44,45]. The present study shows that the values have a trend to increase over time in the 4- and 8-year follow-up groups since the first examination. The MBI provides information about when the break-up of the tear film produces ocular discomfort (nociception activation). Therefore, the increase in temporal values over time may be due to MBI having sensory involvement, and as with symptomatology (Table 1), there being an increased tolerance to discomfort over time [36,46].

Potential damage to the ocular surface was assessed by measuring corneal staining. The results of the present study indicate an increase in corneal staining values between sessions in the 8-year follow-up group (Table 4). The damage to the ocular surface is a consequence of the inflammatory cascade that occurs on the ocular surface due to the instability and hyperosmolarity of the tear film (Table 2 and Table 3) [8]. Therefore, the results obtained may be expected, as the osmolarity values remained elevated throughout the natural course of the disease in the present study. Conversely, DED patients treated with tear substitutes showed a decrease in osmolarity to the normal range, accompanied by an improvement in ocular health with a consequent reduction in corneal staining [42,47,48].

The main strength of this study is that it addresses clinicians’ demand for long-term studies that capture both changes in patients’ symptoms and the characteristic signs of non-adherent DED, which are currently few. Furthermore, in the present study, the natural course of the disease was undertaken not only for one period of time but over three different periods of time since the patient was diagnosed with DED, conducting a follow-up visit at 4, 6 and 8 years. Nevertheless, there are several limitations to consider. First, the possibility that some participants, despite being explicitly instructed to disclose whether they had used artificial tears or other products for managing their DED condition (either regularly or discontinued), may have withheld accurate information and, in fact, used such products. To address this issue in future studies, researchers could include objective diagnostic tools or biomarkers to verify treatment uptake and minimise reliance on self-report. Second, the inclusion of a control group of non-DED or healthy patients would have been valuable to determine whether the observed changes in parameters are solely due to the natural course of the disease or partly related to ageing. It would also be interesting to compare these findings with a group of patients who adhered to the recommended treatment; however, this comparison is challenging due to the wide variability in therapeutic approaches currently available for DED management. Third, the sample size was not the same in the groups and tests, in particular for Group 2, since some patients had not undergone certain diagnostic tests in the first session. Fourth, the present study did not perform a secondary classification of participants according to DED subtype (evaporative dry eye versus aqueous deficient dry eye) or severity (moderate, mild or severe); this could be an interesting factor to consider in future studies to determine the natural course of the specific signs of each DED subtype. Fifth, although inclusion criteria partially accounted for systemic and ocular conditions and concurrent medications, a more extensive collection of demographic and clinical variables would be useful in future studies to control for potential confounding factors. Overall, future research should aim to clarify why symptoms and clinical signs sometimes diverge in DED, by using well-characterised cohorts, verifying treatment adherence, including appropriate controls, scheduling measurements consistently, evaluating neuropathic ocular pain and accounting for disease subtype and severity.

## 5. Conclusions

In conclusion, the present long-term results show that inter-eye osmolarity and FBUT remain stable during the natural course of DED in non-adherent patients, while ocular surface damage (corneal staining) increases over time. However, the subjective symptomatology and the nociceptive blink reflex due to ocular discomfort in these patients decrease since the first examination, which might suggest a possible hypoalgesic response induced by continuous exposure to an unstable environment. Further studies including quantitative sensory testing are warranted to confirm this hypothesis. These insights could spark future research into more effective treatments for DED, focusing on enhancing tear stability and reducing hyperosmolarity to prevent corneal surface damage during the course of the disease. These findings highlight the importance of longitudinal analysis in understanding the changes and progression of ocular surface characteristics over time. Further research and follow-up examinations are necessary to gain a deeper understanding of the underlying mechanisms and implications of these observed differences.

## Figures and Tables

**Figure 1 life-15-01783-f001:**
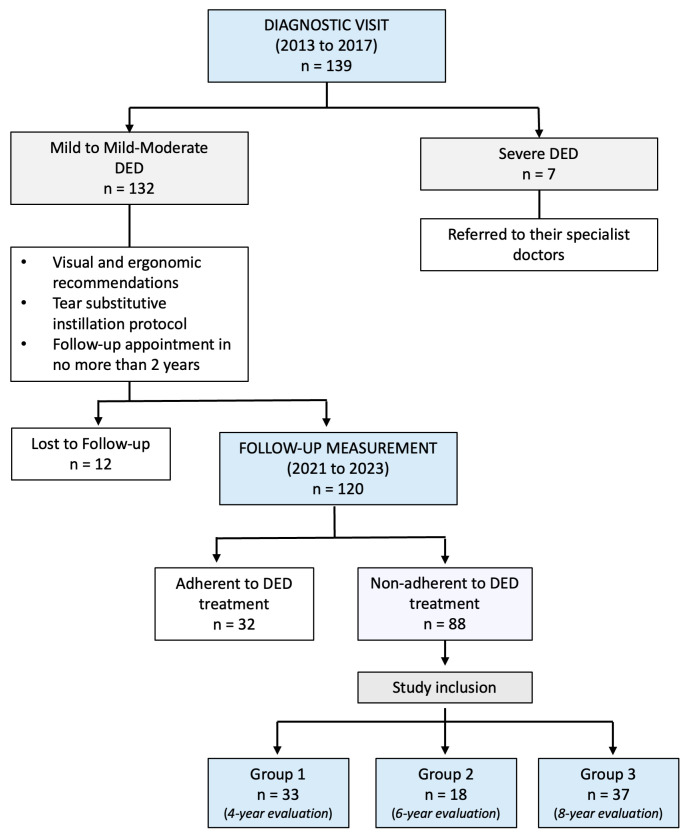
CONSORT-style flowchart of the screened, excluded and included participants.

**Table 1 life-15-01783-t001:** Descriptive statistics and analysis of the differences for the OSDI test scores between sessions in the Group 1 (*n* = 33), Group 2 (*n* = 18) and Group 3 (*n* = 37). *p*-values were determined by paired *t*-test for parametric parameters. * Statistically significant (*p* < 0.05). OSDI = Ocular Surface Disease Index; SD = Standard Deviation; IQR = Interquartile Range.

Parameter	Group	*n* Sample	Session	Mean ± SD	Median (IQR)	Mean Difference ± SD	*p*
OSDI	Group 1 (4 years)	33	Session 1	25.49 ± 16.86	21.67 (10.65–38.35)	−7.61 ± 14.37	<0.001 *
			Session 2	17.79 ± 15.03	13.07 (9.64–24.48)		
	Group 2 (6 years)	18	Session 1	30.37 ± 13.28	32.16 (22.73–42.50)	6.71 ± 19.63	0.058
			Session 2	37.08 ± 22.63	36.46 (16.67–52.08)		
	Group 3 (8 years)	37	Session 1	27.42 ± 15.30	25.00 (16.67–41.67)	−6.07 ± 13.63	<0.001 *
			Session 2	21.36 ± 10.95	20.83 (13.78–27.13)		

**Table 2 life-15-01783-t002:** Descriptive statistics and analysis of the differences for the osmolarity and the osmolarity|OD-OS| values in mOsm/L between sessions in the Group 1 (*n* = 34), Group 2 (*n* = 24) and Group 3 (*n* = 45). *p*-values were determined by paired-t test for parametric parameters. * Statistically significant (*p* < 0.05). Osmolarity|OD-OS| = Osmolarity difference between eyes; SD = Standard Deviation; IQR = Interquartile Range.

Parameter	Group	*n* Sample	Session	Mean ± SD	Median (IQR)	Mean Difference ± SD	*p*
Osmolarity	Group 1 (4 years)	34	Session 1	321.47 ± 14.37	322.00 (311.50–330.00)	8.59 ± 30.16	0.106
			Session 2	330.06 ± 25.58	322.00 (312.00–343.25)		
	Group 2 (6 years)	24	Session 1	316.48 ± 13.87	317.50 (307.00–329.38)	8.44 ± 19.80	0.048 *
			Session 2	324.92 ± 19.27	324.50 (309.75–339.25)		
	Group 3 (8 years)	45	Session 1	315.63 ± 13.03	315.00 (308.00–321.00)	4.34 ± 20.84	0.169
			Session 2	319.98 ± 17.84	320.00 (309.00–329.50)		
Osmolarity |OD-OS|	Group 1 (4 years)	13	Session 1	11.69 ± 10.21	10.00 (6.00–14.50)	1.85 ± 18.24	0.722
			Session 2	13.54 ± 12.14	11.00 (3.00–24.50)		
	Group 2 (6 years)	12	Session 1	14.13 ± 19.26	10.00 (2.00- 14.50)	−2.96 ± 15.65	0.526
			Session 2	11.17± 8.71	9.50 (4.00–18.75)		
	Group 3 (8 years)	14	Session 1	9.43 ± 9.29	7.00 (3.00–12.75)	6.79 ± 14.14	0.096
			Session 2	16.21 ± 13.42	12.00 (6.00–27.00)		

**Table 3 life-15-01783-t003:** Descriptive statistics and analysis of the differences for the mean FBUT and MBI in seconds between sessions in the Group 1 (*n* = 50), Group 2 (*n* = 26) and Group 3 (*n* = 56). *p*-values were determined by Wilcoxon test for nonparametric parameters. * Statistically significant (*p* < 0.05). FBUT = Fluorescein Break-up Time; MBI = Maximum Blink Interval; SD = Standard Deviation.

Parameter	Group	*n* Sample	Session	Mean ± SD	Median (IQR)	Mean Difference ± SD	*p*
FBUT	Group 1 (4 years)	50	Session 1	5.89 ± 5.44	3.94 (1.78–7.42)	0.67 ± 6.87	0.133
			Session 2	6.55 ± 4.15	5.67 (3.62–8.04)		
	Group 2 (6 years)	26	Session 1	10.96 ± 7.26	9.55 (5.17–16.13)	−0.26 ± 7.45	0.751
			Session 2	10.70 ± 7.75	8.38 (5.94–14.05)		
	Group 3 (8 years)	56	Session 1	5.86 ± 5.10	4.36 (2.77–7.60)	0.93 ± 5.47	0.212
			Session 2	6.78 ± 5.05	5.29 (3.94–7.56)		
MBI	Group 1 (4 years)	50	Session 1	10.42 ± 8.68	7.12 (3.71–14.57)	4.88 ± 8.60	<0.001 *
			Session 2	15.30 ± 7.61	12.88 (9.81–21.18)		
	Group 2 (6 years)	26	Session 1	15.98 ± 8.71	13.84 (9.60–21.50)	2.73 ± 8.78	0.101
			Session 2	18.70 ± 9.10	17.15 (11.23–28.00)		
	Group 3 (8 years)	56	Session 1	9.59 ± 6.01	8.57 (6.39–9.87)	4.56 ± 6.24	<0.001 *
			Session 2	14.15 ± 7.05	12.90 (8.72–17.61)		

**Table 4 life-15-01783-t004:** Descriptive statistics and analysis of the differences for the corneal staining scores following the Oxford Scheme between sessions in Group 1 (*n* = 48), Group 2 (*n* = 28) and Group 3 (*n* = 55). *p*-values were determined by Wilcoxon test for nonparametric parameters. * Statistically significant (*p* < 0.05). SD = Standard Deviation; IQR = Interquartile Range.

Parameter	Group	*n* Sample	Session	Mean ± SD	Median (IQR)	Mean Difference ± SD	*p*
Corneal Staining	Group 1 (4 years)	48	Session 1	0.81 ± 0.98	1.00 (0–1.00)	0.31 ± 1.19	0.091
			Session 2	1.13 ± 1.02	1.00 (0–2.00)		
	Group 2 (6 years)	28	Session 1	0.93 ± 1.15	0.50 (0–2.00)	0.07 ± 0.72	0.593
			Session 2	0.86 ± 1.08	0 (0–2.00)		
	Group 3 (8 years)	55	Session 1	0.42 ± 0.74	0 (0–1.00)	0.50 ± 1.09	0.004 *
			Session 2	0.91 ± 1.13	1 (0–2.00)		

## Data Availability

Data is unavailable due to privacy restrictions.

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
