# Peer review of "Long-Term Study of the Changes in Symptoms and Signs of Dry Eye Disease in Participants Non-Adherent to Treatment"

_life, 2025, doi:10.3390/life15111783_

Round 1
Reviewer 1 Report
Comments and Suggestions for Authors
Authors presents an interesting manuscript entitle “Long-term study of the changes in symptoms and signs of dry eye disease in non-adherent participants” to describe “the changes that may occur, both symptoms and signs in non-adherent Dry Eye Disease (DED) patients through periods of 4, 6, and 8 years”. This is an interesting topic with high interest to Eye Care Practitioners than managed DED patients with high potential to be published in Life journal.
This manuscript is well written and presented with a relevant information and is suitable to be published in Life journal. This Reviewer has some minor comments to improve this manuscript to facilitate future readers a better understanding of this manuscript and encourages authors to address these issues previous to publish this manuscript in Life journal.
The strengths of the article: Report of a large follow-up results in untreated dry eye patients to describe long-term clinical variation in symptoms and signs that present high novelty information about natural dry eye disease evolution.
The weaknesses of the article: Some details of the manuscript presentation that are noted as minor changes can be improved to clarify information and results presentation. For example, results presentation could highlight if patients have improved or worse (not just if differences are (or not) statistically significant), this could be important to future clinical application of results of this interesting study.
This reviewer apologizes if he missed any important points or misinterpreted the goals of the study.
Main comments
Study aims description
Authors describe study aim in Lines 78-79 as “the purpose of the present study was to analyse the changes, both symptoms and signs, which may occur in non-adherent DED patients through a period of 4, 6 and 8 years, respectively” that include a detail information about Method (study groups). However, this reviewer guess that study aim could be clearest if is summarized and presented as “The purpose of this study was to assess differences in clinical symptoms and signs of DED in untreated patients to describe long-term disease progression”. After this definition, Material & Method section clearly presents how study was conducted.
Study aim must present in Abstract section in same way that it is showed at the end of Introduction section.
Also, study aim could be clearest if different manuscript sections heading in Results section are rewritten. For example, heading “3.1 Analysis of the symptomatology between sessions” could be clearest if is presented as “3.1 Natural evolution of symptomatology (OSDI score) over time”, or heading “3.2 Analysis of the tear film osmolarity between sessions” by “3.2 Natural evolution of tear film osmolarity over time”, or heading “3.3 Analysis of the tear film stability between sessions” by “3.3 Natural evolution of the tear film stability over time” and heading “3.4 Analysis of the corneal staining between sessions” by “3.4 Natural evolution of the corneal staining over time”. Highlighting in these heading that differences are related with natural evolution over time, better match between study aim and methodology is obtained comparing with current proposed presentation focused in describe the differences between sessions (that reader need to remember that are baseline and after 4, 6 and 8 years).
Material and Methods description could be improved with minor recommendations (see comments below).
Results presentation
This Reviewer recommends that Authors include in result section clinical meaning of the results, that is if results improve o worse symptoms or signs with time (not just difference value and statistical significance). For example, in Line 229-234, Authors present differences in OSDI score. This difference will be clearest if this paragraph is modified, for example “OSDI score slight improve after 4 years of DED diagnosis (Group 1, difference close 12 points, P<0.001) and after 8 years of diagnosis (Group 3, difference of 9 points, P<0.001), but after 6 years (Group 2, score was stable, just 3 points of difference, P=0.328). These results suggest that patients become more tolerant with DED symptomatology”. This results presentation facilitates future readers better understanding of these interesting results.
This approach is, also, applicable to all results presentation; tear film osmolarity (lines 240-242), osmolarity OD-OI (lines 244-246), tear film stability (FBUT lines 257-259 and MBI lines 259-261), and corneal staining (lines 272-274) in main Results section and Abstract section, as well.
Discussion and Conclusions
This Reviewer recommends cite the Tables when each result is discussed because Authors have analyzed many parameters, and future readers need to see clear as possible what data are discussing in each moment. For example, Line 300 refer to this study results (but previous paragraphs are referring to other literature reports) so if Authors include Table description (for example “in the present study it was observed that the OSDI scores decreased (symptomatology improvement) significantly over time (Table 1) for all groups except Group 2, which remained stable”, these clarifications -clinically meaning of differences and table citation- will help future readers to better understanding of Discussion section. This approach will improve discussion of all results.
See detailed recommendations of minor changes suggested.
Minor recommendations
Abstract
Line 14-16 “This prospective observational study analyses changes in, both symptoms and signs in Dry Eye Disease (DED) patients over time comparing diagnosis visit with visits at 4, 6, and 8 years without any treatment (non-adherent patients)”.
Line 23 “… were found between diagnostic visit (session 1) and follow up visit (session 2) in ODSI score (improving symptomatology) in Groups 1 and 3…”
Line 25 Typing mistake “p≤0.001),Osmolarity”.
Line 26 Possible typing mistake “13,2 ±19.6, p=0.011”
Line 29-30 Typing mistake (different letter type) “In non-adherent DED patients”
Line 33 Replace “first examination” by “basal” or “diagnostic visit”.
Introduction
Line 51 delete “According to the recent TFOS DEWS III report,”, it is redundant because reference clearly describes where information was obtained.
Line 56-64 In this paragraph the role of tear film hyperosmolarity as trigger or key factor is repeated. Please replace by “Within the signs, a high tear film osmolarity may be responsible for the pathophysiological process in which the disruption of ocular surface homeostasis occurs [7]. Therefore, hyperosmolarity is a key factor being a trigger and main contributor for the instability of the tear film, inflammation, and damage of the ocular surface; that induces the “Vicious Circle” of DED [8,9], because this inflammatory cascade reduces the expression of glycocalyx mucins and induces apoptosis of epithelial cells, decrease ocular surface wettability, increase tear film instability, and ultimately leading to hyperosmolarity”.
Line 65-81 Please revise letter type size (is bigger than previous paragraphs).
Material & Method.
Line 102 replace “basal measurement” by “diagnostic visit”.
Line 103 replace “… confirm their diagnosis …” by “… confirm DED diagnosis …”.
Line 103 replace “… in the centre …” by “.. in the Optometry eye clinic, …”.
Line 106 Please clarify this sentence. Are Authors suggesting that in all of the 100% of patients with suspected DED after the eye exam DED was confirmed? Or that just confirmed DED diagnosis patients were included in this study?
Line 117-118 “During this contact process, the participants were asked if they followed the recommendations provided by eye care practitioners in diagnostic visit”.
Line 148 “None of the participants used any kind of medication, artificial tears or wore contact lenses during the assessed period, after the diagnostic visit.
Line 203 delete “our”, scientific literature is clearest without use of pronouns.
Line 216-221 must be moved after line 209-210 (this is a description of the first statistical analysis to justify statistical comparison described in lines 210-215.
Results
A table to summarize comparison described in lines 223-226 is welcome to improve study results presentation.
Line 252 Table 2 Please revise format to be similar as Table 1.
Line 271-274 Typing mistake. Please revise text format (letter size, paragraph, etc.) to guarantee that text format is like previous paragraphs.
Line 278 Table 4. Please avoid using hyphen to break word “Differ-ence”.
Discussion
Line 291 “… 8 years since the initial diagnosis of DED in non-adherent patients”.
Line 344 “The main strength of this study is that it addresses clinicians' demand for long term studies …”.
Line 348-349 “… conducting a follow-up visit at 4, 6 and 8 years”.
Line 352 Are Authors referring of a “control group of non DED patients”? if answer is yes, this sentence could be clearest if this clarification is included.
Conclusions
Line 365 “… the present long-term results show …”.
Line 366 “… natural course of the DED …” and delete “diagnosed with DED”.
Line 367 could be clearest if is presented as “… while ocular surface damage (corneal staining) increases over time”.
Line 375 “Further research and follow-up examinations are necessary to gain a …”
Author Response
Please find the attached document with the responses.

Reviewer 2 Report
Comments and Suggestions for Authors
This longitudinal prospective study addresses a clinically important and understudied question: how signs and symptoms of dry eye disease (DED) evolve in patients who report not following treatment recommendations over 4, 6, and 8 years. The study follows the recommended TFOS DEWS methodology and provides interesting observations on symptom–sign dissociation, tear osmolarity, blink dynamics, and corneal staining. However, several important methodological, statistical, and interpretive issues limit confidence in the conclusions.
- Study design and participant flow are unclear; participant attrition must be documented.
Provide a CONSORT-style flow diagram that shows numbers screened, eligible, contacted, declined, excluded (with reasons), lost to follow-up, and analysed for each outcome and each timepoint; explicitly state how the initial 120 contacted subjects resulted in 88 analysed (33 + 18 + 37)
- The operational definition of “non-adherent” is insufficient and risks bias from misclassification.
Report the exact wording/questions used during recruitment to categorise non-adherence, note the recall period, and discuss misclassification risk. If possible, supplement the self-report with objective verification (e.g., pharmacy refill records, prescription databases), and report how many participants could not be verified.
- Sample size justification and power calculations are inconsistent and incomplete.
Recalculate post-hoc power for each pre-specified primary outcome using observed variances and present the achieved power per group. Report the originally assumed effect sizes and justify their choice (cite prior studies). Explain why Group 2 is smaller and discuss how this imbalance affects type II error and interpretation.
- Results presentation and tables should be clearer and unambiguous.
In Table 1 and elsewhere, indicate direction of change (follow-up minus baseline) for “Mean Difference ± SD”, standardise decimal places, and for non-parametric variables, present median (IQR) rather than mean ± SD. Add the number of observations per test in each cell (n) to show missing data.
- Inter-examiner and intra-examiner reliability are not reported for key measures.
Provide intra- and inter-observer reliability metrics for osmolarity, FBUT, MBI, and corneal staining measured from the video recordings, and describe calibration procedures and equipment lot control across sessions.
- The hypoalgesia explanation is speculative and should be framed as a hypothesis.
Rephrase claims about decreased nociception as a plausible hypothesis and cite neurosensory literature; recommend future studies include quantitative sensory testing to evaluate neuropathic mechanisms.
- Contradictions in osmolarity statements must be reconciled.
Reconcile the assertion “osmolarity stabilises” with the statistically significant osmolarity increase observed in the 6-year group by (a) reporting effect sizes and clinical relevance for that change, (b) exploring whether the effect is driven by outliers or measurement timing, and (c) considering diurnal/seasonal variability or device lot effects as alternative explanations.
- The absence of a control or comparator group weakens causal inference.
Prominently acknowledge this limitation in the Discussion and, if feasible, plan or append a matched control analysis (adherent DED patients or age-matched healthy controls) using institutional datasets to disentangle ageing effects from disease progression.
- Potential confounders (sex, age, comorbidities, medications, environment) are insufficiently addressed.
Provide baseline demographic/clinical tables that include sex, age, menopausal status, systemic diseases, systemic and ocular medications, occupational screen exposure, contact lens history, and environmental variables; if data are unavailable, list their absence as a study limitation and discuss likely directional biases.
- Discordance between signs and symptoms requires more detailed analysis.
Perform correlation analyses (Spearman or Pearson as appropriate) between OSDI and MBI, OSDI and osmolarity, and MBI and corneal staining for each group; report correlation coefficients and p-values, and discuss potential mechanisms for non-concordance.
- Title, abstract, and keywords could be optimised for precision and discoverability.
Consider a title emphasising the design and population. In the abstract, briefly define “non-adherent” (e.g., “patients who reported not following prescribed dry-eye therapy”), and add keywords such as “longitudinal study”, “tear film osmolarity”, and “maximum blink interval”.
- Explain why some expected temporal correlations were not observed.
In the Discussion, address why temporal correlation between length of follow-up and specific outcomes is inconsistent (e.g., no time-trend for FBUT vs. increases in staining): propose biological explanations (compensatory ocular surface mechanisms), measurement noise, or cohort heterogeneity, and recommend specific future study designs to test these hypotheses.
- Limitations should be expanded and prioritised.
List limitations explicitly and in order of likely impact: (1) self-reported non-adherence without objective verification; (2) lack of control group; (3) unequal group sizes and missing data; (4) absence of DED subtype stratification; (5) incomplete recording of systemic/ocular comorbidities and medications; (6) potential measurement variability due to different examiners; and (7) observational design precluding causal inference.
- Recommendations for future research.
Recommend prospective cohorts with pre-specified adherence verification, inclusion of controls, standardised measurement times, sensory testing for neuropathic pain, and stratification by DED subtype and severity to clarify mechanisms behind symptom–sign dissociation.
If the authors address these methodological and reporting concerns, the study has the potential to provide valuable long-term insight into the natural course of DED in non-adherent patients.
Author Response

(The authors gave the same response as above.)

Reviewer 3 Report
Comments and Suggestions for Authors
Dear authors:
- Could you make a definition about " non-adhrenent" paticipants in 2.1 sample and study "
- The documents between Line 101 and Line 138 are very difficult to understand. May the authors metion these contents simply ?
Author Response

(The authors gave the same response as above.)

Round 2
Reviewer 2 Report
Comments and Suggestions for Authors
All suggestions were suitably considered, the requested changes were implemented, and all explanations were logical and grounded. This has resulted in an improved manuscript. Therefore, the manuscript can be accepted in its current form.
Reviewer 3 Report
Comments and Suggestions for Authors
Thanks for your revision.